# Corn Flour Intake, Aflatoxin B_1_ Exposure, and Risk of Esophageal Precancerous Lesions in a High-Risk Area of Huai’an, China: A Case-Control Study

**DOI:** 10.3390/toxins12050299

**Published:** 2020-05-06

**Authors:** Shaokang Wang, Da Pan, Ting Zhang, Ming Su, Guiju Sun, Jie Wei, Ziqi Guo, Kai Wang, Guang Song, Qingyang Yan

**Affiliations:** 1Key Laboratory of Environmental Medicine and Engineering of Ministry of Education, Department of Nutrition and Food Hygiene, School of Public Health, Southeast University, Nanjing 210009, China; pantianqi92@foxmail.com (D.P.); zhangtingzg@foxmail.com (T.Z.); gjsun@seu.edu.cn (G.S.); jiewei1997@foxmail.com (J.W.); ziqiguo97@126.com (Z.G.); 2Huai’an District Center for Disease Control and Prevention, Huai’an 223200, China; sumingcdc@protonmail.com (M.S.); wangkai899@126.com (K.W.); songguangcdc@protonmail.com (G.S.); yanqingyang_cdc@126.com (Q.Y.)

**Keywords:** Aflatoxin B_1_ exposure, duplicate diet, serum biomarker, corn flour intake, esophageal precancerous lesion

## Abstract

Aflatoxin B_1_ (AFB_1_), which has potent toxicity and carcinogenicity, is a common contaminant of important agricultural commodities. This study aimed to investigate the frequency of corn flour intake and assess the exposure to AFB_1_ via direct detection of AFB_1_ in the diet and serum AFB_1_ exposure biomarker, so as to evaluate their associations with the risk of esophageal precancerous lesions (EPL). A case-control study based on three-day duplicate diet samples was performed in Huai’an District. One hundred EPL cases and 100 healthy controls were enrolled and required to be age- (±2 years) and gender-matched. The concentration of AFB_1_ in food samples and the level of serum AFB_1_-albumin (AFB_1_-Alb) adduct were quantitatively analyzed. Results showed that corn flour intake was positively associated with serum AFB_1_-Alb adduct level (*p* for trend = 0.003), dietary AFB_1_ exposure (*p* for trend < 0.001), and the risk of EPL (*p* for trend = 0.017). Increased serum AFB_1_-Alb adduct level was associated with an increased risk of EPL as well (*p* for trend < 0.001). In conclusion, corn flour may be an essential source of AFB_1_ in Huai’an District, whereas high exposure to AFB_1_ is likely to be an important risk factor contributing to the progression of EPL.

## 1. Introduction

Esophageal cancer (EC) is the sixth leading cause of death from cancer in the world, caused by many risk factors which differ by histological type, population, and region [1]. Esophageal squamous cell carcinoma (ESCC), the predominant histological sub-type of EC in developing countries including China, accounts for about 90% cases of EC each year [2]. The recognized esophageal precancerous lesions (EPL) for ESCC can be classified into mild, moderate, and severe esophageal squamous dysplasia, which have been found to be associated with approximately a 3-, 10-, and 30-fold higher risk of ESCC than normal [2,3,4]. It was reported that 24% of mild dysplasia, 50% of moderate dysplasia, and 74% of severe dysplasia would develop ESCC during a 3.5-year period [4]. The primary risk factor that increases the risk of ESCC/EPL in one population may not be significantly associated with this cancer in another, which means that there may be completely different patterns of epidemiology between populations, and the geographic distribution of ESCC/EPL varies greatly [1,5]. In Huai’an District, an endemic region of EC in the Jiangsu Province of East China (Figure 1), the crude incidence and mortality of EC during 2008 and 2012 were 96.15/100,000 and 63.25/100,000, respectively [5]. Our previous study in this region reported that a distinct epidemiological pattern of EPL was observed here: the use of alcohol and tobacco, which are the established risk factors, plays only a minor role in the risk of EPL, whereas the factors which influence both genders equally, such as environmental exposures and dietary factors, are found to take the main responsibility in the process of carcinogenesis of ESCC [5].

Aflatoxins, secondary metabolites with high toxicity produced by *Aspergillusflavus* and *Aspergillus parasiticus*, are common dietary contaminants and have emerged as a global public health concern owing to their toxic effects on human and animals [6,7]. In 2007, the European Food Safety Authority (EFSA) assessed the possibility of a potential increase in consumers’ health risks if higher levels of aflatoxins were permitted for almonds, hazelnuts, and pistachios. The panel also concluded that exposure to aflatoxins from all food sources should be kept as low as reasonably achievable because aflatoxins are genotoxic and carcinogenic [8]. At the same time, the public health and food safety areas were subjected a significant legislative pressure with a large amount of passed European acts, including the regulations on the risks represented by mycotoxins [9]. All the characteristics make regulations extremely efficient work tools, with widespread use in regulating the most varied subdomains, from animal health issues to contaminants and mycotoxins [10].

Among the known aflatoxin derivatives, aflatoxin B_1_ (AFB_1_) is known to have the strongest toxicity and carcinogenicity [11,12,13]. In peripheral blood, AFB_1_-albumin (AFB_1_-Alb) adduct has been considered the most reliable biomarker for a biologically effective dose of exposure to AFB_1_ in humans because it has a relatively long half-life, reflects AFB_1_ exposure over several months, and keeps stable during a long-term deep-frozen storage [14,15,16,17]. Corn, corn products, and other cereals are commonly contaminated with aflatoxins in many developing countries, especially in tropical and subtropical regions [7,13,18]. In Huai’an District, corn flour is consumed as one of the major staple foods, and its material, corns are basically dehydrated by exposure to sun after harvest for storage [5]. However, aflatoxins often develop in stored corns and flours, and therefore result in a health hazard to residents. Our previous studies reported that high consumptions of corn and corn flour were associated with increased risk of EPL in Huai’an District [5], and the concentration of AFB_1_ in corn samples in Huai’an District (median, 13.5 μg/kg (1.2–136.8)) was significantly higher than that in Huantai County (median, 1.3 μg/kg (0.4–2.2)), which is a low-risk area for EC [19].

To date, most of the previous studies were focused on the relation between AFB_1_ exposure and hepatocellular carcinoma, but there was limited epidemiological evidence supporting the association between AFB_1_ exposure and ESCC. Our recent study reported that AFB_1_ exposure in serum was associated with increased risk of ESCC in Huai’an [20], whereas it remains unclear whether AFB_1_ also contributes to the earlier stage of ESCC progression, i.e., EPL. In a number of studies, duplicate diet study has been suitably used to accurately evaluate the dietary intake of nutrients and contaminants by capturing exposures from food as prepared and eaten, thus is recognized as the approach providing the best data for overall dietary exposure assessment [21,22]. Here, a case-control study based on three-day duplicate diet samples was performed in Huai’an District, aimed to evaluate the exact value of dietary AFB_1_ exposure, serum AFB_1_-Alb adduct level and investigate the frequency of corn flour intake in both healthy controls and EPL cases, so as to elucidate the relationships between corn flour intake and AFB_1_ exposure, and between AFB_1_ exposure and the risk of EPL in this rural region.

## 2. Results

### 2.1. Socio-Demographic Characteristics of the Subjects

In this study, 100 EPL cases and 100 healthy controls with a mean age of 64.45 ± 5.34 and 64.38 ± 5.09 years were enrolled and required to be age- (±2 years) and gender-matched. In all, 52 pairs were males and 48 pairs were females. The average body mass index (BMI) of EPL cases and healthy controls were 23.72 ± 3.21 kg/m^2^ and 23.75 ± 3.32 kg/m^2^. Two independent samples *t*-test indicated that there was no statistical significance between the two groups in age and BMI (*p* > 0.05).

### 2.2. Serum AFB_1_-Alb Adduct Level and Dietary AFB_1_ Exposure of the Subjects

The distribution of serum AFB_1_-Alb adduct level in EPL cases and healthy controls is shown in Table 1 and Figure 2. Wilcoxon signed-rank test indicated that serum AFB_1_-Alb adduct level in EPL cases was significantly higher than that in healthy controls (*p* < 0.001). However, there was no statistically significant difference in dietary AFB_1_ exposure between the two groups (*p* = 0.891). Figure 3 illustrates that there would be a weak positive correlation between dietary AFB_1_ exposure and serum AFB_1_-Alb adduct level (Pearson correlation coefficient = 0.269, *p* < 0.001).

### 2.3. Association between AFB_1_-Related Variables and Risk of EPL

Serum AFB_1_-Alb adduct level and dietary AFB_1_ exposure were categorized into tertiles. Results of regression models were expressed by calculated odds ratio (OR) and 95% confidence interval (CI). As shown in Table 2, adjusted results indicated that the second (OR = 8.11, 95% CI 2.56–25.71) and third tertiles (OR = 25.12, 95% CI 7.29–87.80) of serum AFB_1_-Alb adduct level were positively associated with the risk of EPL when compared with the first tertile (*p* for trend < 0.001). Increased dietary AFB_1_ exposure was associated with a non-significantly increased risk of EPL (*p* for trend = 0.648). Increased frequency of corn flour intake was found to be associated with an increased risk of EPL (*p* for trend = 0.017), and consuming corn flour four times a week or more may confer over threefold increase in risk of EPL relative to the frequency of less than once a month. In addition, subjects whose stored grains were found to be mildewed may have tenfold increase in risk of EPL (*p* = 0.020).

### 2.4. Association between Frequency of Corn Flour Intake and AFB_1_-Related Variables

Serum AFB_1_-Alb adduct level and dietary AFB_1_ exposure were categorized into binary classifications of high and low levels according to their median values. As shown in Table 3, binary logistic regression analysis indicated that frequency of corn flour intake was positively associated with both serum AFB_1_-Alb adduct level (*p* for trend = 0.003) and dietary AFB_1_ exposure (*p* for trend < 0.001).

## 3. Discussion

In the present study, we assessed the exposure to AFB_1_ using corresponding exposure biomarkers in serum and diet samples, investigated the frequency of corn flour intake, and evaluated their associations with the risk of EPL in Huai’an District. Based on the results, the frequency of corn flour intake was positively associated with both serum AFB_1_-Alb adduct level and dietary AFB_1_ exposure. Additionally, increased level of serum AFB_1_-Alb adduct, higher frequency of corn flour intake and mildew of stored grains were significantly associated with increased risk of EPL.

Although AFB_1_ is more commonly related to the risk of the hepatocellular carcinoma [23], some studies have suggested its potential link to the risk of ESCC in Huai’an [20], as well as its interaction with fumonisin B_1_ on ESCC risk in high-risk areas including Cixian, Linxian and Huai’an [19,24]. In Africa, parts of Asia, and Latin America, many poor people are exposed to mycotoxins such as aflatoxins and fumonisins on a daily basis by eating their staple diet of groundnuts, corn, and other cereals [23]. For example, a study in the Golestan province of Iran reported that the levels of total aflatoxins and AFB_1_ in wheat flour samples were significantly higher in high-risk areas rather than the low-risk areas [25]. Similarly, the Huai’an rural region is generally lower middle class where many residents consume corn flour as the major staple food. The present study, which assessed the exposure to AFB_1_ via direct detection of AFB_1_ in diet samples and serum AFB_1_ exposure biomarker, confirmed the positive relationship between corn flour intake and AFB_1_ exposure in both diet and serum, and indicated that increased intake of corn flour and elevated level of serum AFB_1_-Alb adduct were associated with a higher risk of EPL. These findings were consistent with previous studies which assessed the associations among corn flour intake, AFB_1_ exposure, and risk of EPL or ESCC in Huai’an [5,19,20].

However, it should be noted that although there was a weak positive correlation between dietary AFB_1_ exposure and serum AFB_1_-Alb adduct level (Pearson correlation coefficient = 0.269, *p* < 0.001), the positive relationship between dietary AFB_1_ exposure and the risk of EPL was not statistically significant (*p* for trend = 0.648). Therefore, it is possible that some residents are more susceptible to AFB_1_, which influences the metabolism of AFB_1_ in vivo and contributes to a relatively high level of serum AFB_1_ exposure biomarker and a higher risk of EPL. For example, there may be relationships between AFB_1_ exposure, the enzymatic capacity (i.e., P450 enzyme) to generate putative reactive AFB_1_ metabolites, and the formation of DNA or protein adducts and metabolites in individuals [26]. Additionally, gene regulation may play an important role in the susceptibility as well, and the mutagenesis by AFB_1_ may depend on the context of DNA [27]. The mutations single-nucleotide polymorphism and aberrant hypermethylation of tumor-suppressor gene p53 have been found in esophageal carcinogenesis, resulting in the loss of its function to inhibit malignant transformation [28,29,30,31]. The presence of the R246S transgene in transgenic mice, which is the equivalent of the human R249S p53 mutant, was found to increase and accelerate the incidence of higher-grade liver cancer upon AFB_1_ exposure [32]. By using transgenic mice models (Hupki mice, human p53 knock-in), a previous study found that without any mutations in p53, Hupki mice were more susceptible to AFB_1_ exposure and subsequent formation of cancer than the mice with wild-type murine p53, thus there would be other ways of inactivation of p53 found in human exposure to AFB_1_ that may accelerate the transition of carcinoma [33]. For instance, hypermethylation of p53 promoter, which is an epigenetic change associated with silencing of protein expression of the gene, was commonly found in EPL/ESCC cases in Huai’an District [28]. Further studies on cancers and interaction between gene regulation and AFB_1_ exposure will be of great significance.

To the best of our knowledge, the development and progression of ESCC is a complex, multi-step continuous dynamic process, often caused by the cooperation of multiple carcinogenic factors. However, the primary risk factor that increases the risk of ESCC/EPL in one population may not be significantly associated with this cancer in another. As reported in previous studies, alcohol use and tobacco smoking, which are the established risk factors for ESCC in other areas, play only a minor role in Huai’an District, whereas the factors such as low consumption of fruits and vegetables, frequent consumption of high-temperature food or beverages, high dietary exposure to nitrite, deficiency of certain nutrients, genetic polymorphism, as well as the exposure to mycotoxins including AFB_1_, have been found to play important roles in the process of carcinogenesis of ESCC in this region [5,19,20,28,34,35,36,37,38,39]. Further studies will be needed to confirm these findings, explore other potential factors and evaluate the possible interaction among the established factors on ESCC development in the studied population in Huai’an District so that the prevention strategies for ESCC/EPL can be improved. The limitation of the present study is that because of some restrictions and difficulties in the implementation of duplicate diet study, only 200 subjects were included, and diet samples were only collected between July and August. Therefore, the number of subjects for the analysis of mildew of stored grains was not sufficient, and the data may not reflect the dietary AFB_1_ exposure during other seasons.

## 4. Conclusions

Corn flour intake was positively associated with serum biomarker for AFB_1_ (AFB_1_-Alb adduct level), dietary AFB_1_ exposure, and the risk of EPL, while the increased level of serum AFB_1_-Alb adduct, and mildew of stored grains were significantly associated with the increased risk of EPL. Therefore, it can be concluded that corn flour is likely to be an essential source of AFB_1_ in Huai’an District due to the suitable environment for fungal growth during the grain storage, whereas high exposure to AFB_1_ may be an important risk factor of EPL, suggesting that AFB_1_ also contributes to the earlier stage of ESCC progression. In addition, some individuals may be more susceptible to AFB_1_ in some cases, such as enzymatic capacity and gene regulation which may affect the metabolism of AFB_1_ in vivo, contribute to the increased level of serum AFB_1_ exposure biomarker, and result in the increased risk of tumor initiation. Thus, it is essential to establish cost-effective prevention strategies to prevent the AFB_1_contamination in foods and reduce the intake of contaminated foods in this region.

## 5. Materials and Methods

### 5.1. Study Population

In order to enhance the prevention of ESCC in Huai’an, the government and Cancer Foundation of China have supported the Early Diagnosis and Early Treatment Project of Esophageal Cancer (EDETPEC) in the endemic region since 2010 [5,28]. As described in our previous study [34], more than 2000 residents aged from 35 to 75 years old who had lived in the towns Jiqiao, Jingkou, and Qiuqiao for more than five years were recruited and underwent a routine endoscopy examination in the EDETPEC between January 2015 and June 2017. A total of 100 EPL cases and 100 healthy controls were selected from this population and matched by gender, age (±2 years) and villages where they lived. Only mild and moderate esophageal squamous dysplasia were included in the EPL cases, as the residents diagnosed with severe esophageal squamous dysplasia or carcinoma-in-situ were asked to take medical treatment. In addition, individuals with cancer history, cardiovascular and cerebrovasculardiseases, Alzheimer’sdisease, schizophrenia, mobility problems, and other diseases that may affect their diet were excluded. The study protocol was approved by the Institutional Review Board of Southeast University Zhongda Hospital (Approval No.: 2012ZDllKY19.0; Approval date: 10 September 2012; and Approval No.: 2016ZDKYSB017; Approval date: 4 March 2016), in accordance with the Declaration of Helsinki. All the participants signed written informed consent.

### 5.2. EPL Diagnosis

As described previously [5,28], subjects were required to undergo a routine endoscopy examination. Esophageal mucosa was stained with 1.2% Lugol’s iodine solution and then observed. Normal esophageal mucosa would turn brown (iodine-positive), whereas dysplastic lesions would remain unstained (iodine-negative). The unstained tissues were sampled, and biopsies were oriented on filter paper, placed in 10% phosphate-buffered formalin and transferred to the pathology laboratory. The biopsies were processed to paraffin blocks, prepared on the slides, and then stained with hematoxylin-eosin for histopathological analysis. The abnormalities including the presence of nuclear atypia, loss of normal cell polarity and abnormal tissue maturation without the invasion of epithelial cells through the basement membrane, are confined to the lower third of the epithelium in mild dysplasia, are present in the lower two-thirds of the epithelium in moderate dysplasia, and also involve the upper third of the epithelium in severe dysplasia [40].

### 5.3. Sample Collection

Epidemiological data of socio-demographics, lifestyle, and eating habits were collected by face-to-face interviews using a questionnaire, and a validated qualitative food frequency questionnaire was used to estimate the dietary intake. Fasting blood samples were collected and centrifuged at 3000 rpm for 5 min to obtain separated serum and leukocyte, then they were stored in the −80 °C refrigerator immediately. From July to August 2017, duplicate diet samples of all foods and beverages (breakfast, lunch, dinner, and any other snacks) consumed daily were collected and recorded by well-trained investigators in three consecutive days [34]. The three consecutive days included two working days and a weekend day. Inedible parts were removed, and the weight and quality of the collected food samples were consistent with the actual food consumption of subjects.

The collected whole day food samples were delivered to the local health station by investigators every day. Then the total amount of food for each subject was weighed and pooled to obtain a composite sample. The composite samples were homogenized using a homogenizer to obtain 600 homogenized samples, which represent 200 subjects. Two percent weight of three homogenized samples for each subject were taken and mixed to obtain 200 three-day diet samples for the test. These samples were stored at −20 °C until analysis.

### 5.4. Determination of Dietary AFB_1_Exposure and Serum AFB_1_-Alb Adduct Level

According to the standard of determination of AFB_1_ in foods (GB/T 5009.22-2003), 5.0 g of a three-day diet sample, and 25 mL of aqueous methanol solution (50% *v*/*v*) were mixed in a 100 mL conical flask and then filtered after shaking for 15 min. The initial volumes of the filtrate were discarded, and the remaining filtrate was prepared for the test. Then the concentration of AFB_1_ in the food sample was determined by double-antibody-sandwich enzyme-linked immunosorbent assay (ELISA) using the AFB_1_ ELISA kit (Jiangsu Suwei Microbiology Research Co., Ltd. Wuxi, China). Serum AFB_1_-Alb adduct level was determined by using the AFB_1_-Alb ELISA kit (Senbeijia Biotechnology Co., Ltd. Nanjing, China). All operations were in strict accordance with manufacturers’ instructions. Optical density (OD) values at a wavelength of 450 nm were measured using a microplate spectrophotometer (Epoch, BioTek Instruments, Winooski, VT, USA), and then the concentrations of AFB_1_ in diet sample and serum AFB_1_-Alb adduct were calculated accordingly. Additionally, daily dietary AFB_1_ exposure was calculated as follows:(1)$$X=\frac{C\times V\times m_{0}}{m_{1}\times 3}$$

In the equation, *X* means daily dietary AFB_1_ exposure (μg/d); *C* means the concentration of AFB_1_ in food calculated from the above assay (μg/mL); *V* means the volume of filtrate (25 mL); *m*_0_ means the total weight of three-day diet sample (g); *m*_1_ means the weight of sample used for assay (5.0 g).

### 5.5. Statistical Analysis

Data collected from the questionnaire were double-entered and validated in an established database with Epidata version 3.1 (EpiData Association, Odense, Denmark) and then processed in Microsoft Excel. Statistical analyses were conducted using SPSS version 22.0 (SPSS, Chicago, IL, USA). Two independent samples *t*-test and Wilcoxon signed-rank test were performed to evaluate differences in general characteristics, serum AFB_1_-Alb adduct level, and dietary AFB_1_ exposure between EPL cases and healthy controls, wherever appropriate. Bivariate correlation analysis was conducted to assess the correlation between serum AFB_1_-Alb adduct level and dietary AFB_1_ exposure. The continuous variables of serum AFB_1_-Alb adduct level and dietary AFB_1_ exposure were categorized into tertiles (Tertile 1, Tertile 2, Tertile 3) or binary classifications of high and low levels by median according to different analysis purposes. Conditional logistic regression (equivalent to stratified Cox proportional hazards regression) and unconditional binary logistic regression analyses were performed to assess the associations between AFB_1_-related variables and risk of EPL, and between corn flour intake and AFB_1_-related variables, respectively. Analyses were adjusted for confounding variables including gender, age, BMI, education level, annual income, tobacco smoking, and alcohol drinking. Tests for a linear trend were performed by assigning the median value of each category of detailed data of variables as a continuous variable in the models. The obtained *p* for trend is used to test whether there is a certain linear trend between the independent and dependent variables. Results of regression models were expressed by calculated odds ratio (OR) and 95% confidence interval (CI), where the OR represents the odds that an outcome will occur given a particular exposure when compared to the odds of the outcome occurring in the absence of the exposure (OR > 1, exposure is associated with higher odds of outcome; OR < 1, exposure is associated with lower odds of outcome; OR = 1, exposure does not influence odds of outcome), and the 95% CI represents the precision of the OR (in practice, the 95% CI is a proxy for the presence of statistical significance if it does not overlap the null value, e.g., OR = 1) [41]. Statistical significance for all tests was considered as *p* value < 0.05 (two-tailed).

## Figures and Tables

**Figure 1 toxins-12-00299-f001:**
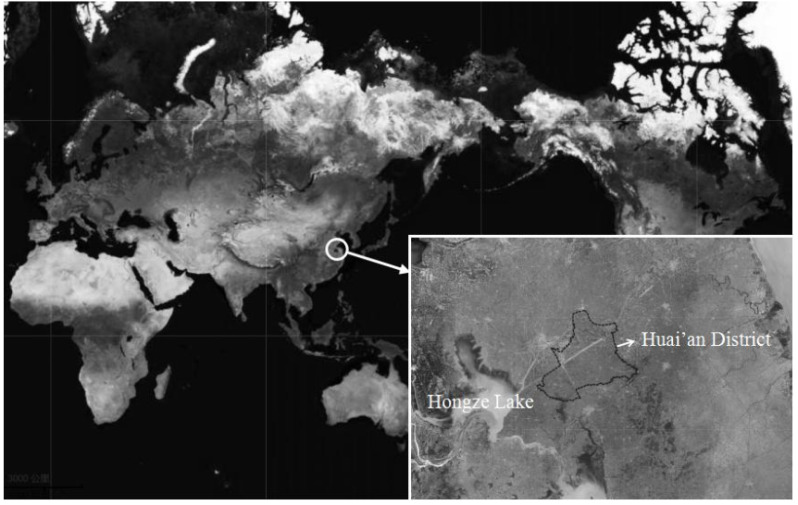
Location of Huai’an District in China. Reprinted from Map of Huai’an District by Bajiu internet services, 29 March 2020, retrieved from https://bajiu.cn/ditu/?qh=20671. Copyright 2006–2020 by Bajiu internet services. Texts and indicators were added with Microsoft Word 2007 (Microsoft Inc., Redmond, WA, USA) (https://www.microsoft.com/en-us/).

**Figure 2 toxins-12-00299-f002:**
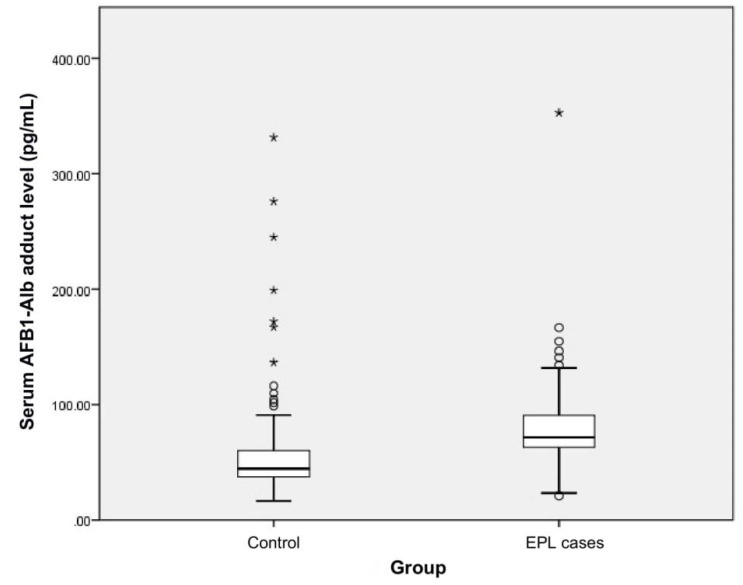
Box and whiskers plot of serum AFB_1_-Alb adduct level in EPL cases and healthy controls. The box values range from 25 to 75 percentiles. The line within the box represents the median. The T-shaped bars at both sides of the box represent 5 and 95 percentiles of data. The dots and asterisks represent outliers and extremes, respectively.

**Figure 3 toxins-12-00299-f003:**
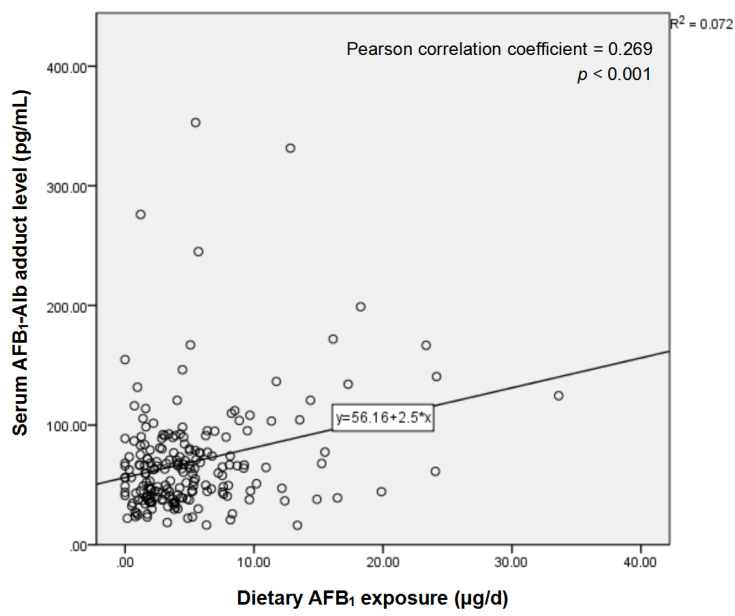
Scatter plot of correlation between dietary AFB_1_ exposure and serum AFB_1_-Alb adduct level. Each dot’s horizontal position indicates a subject’s dietary AFB_1_ exposure (in μg/d) and the vertical position indicates the subject’s serum AFB_1_-Alb adduct level (in pg/mL).

**Table 1 toxins-12-00299-t001:** AFB_1_-Alb adduct level and dietary AFB_1_ exposure of study subjects (Median (25th–75th)).

AFB_1_-Related Varibles	Control (n = 100)	EPL (n = 100)	*p* Value *
**Serum AFB_1_-Alb adduct level (pg/mL)**	44.48 (36.88 < 59.95)	71.40 (62.80–91.07)	<0.001
**Dietary AFB_1_ exposure (μg/d)**	3.47 (1.86–7.46)	4.19 (1.74–6.15)	0.891

* *p* value of Wilcoxon signed-rank test.

**Table 2 toxins-12-00299-t002:** ORs (95% CIs) for AFB_1_-related variables with EPL.

AFB_1_-Related Varibles	No. of Cases/Controls	Crude OR (95% CI)	*p* Value	Adjusted OR (95% CI) *	*p* Value
**Serum AFB_1_-Alb adduct level (pg/mL)**					
Tertile 1 (16.25–45.00)	14/54	1.00 (reference)	–	1.00 (reference)	–
Tertile 2 (45.01–71.32)	36/30	5.05 (2.01–2.70)	0.001	8.11 (2.56–25.71)	<0.001
Tertile 3 (71.33–352.85)	50/16	11.69 (4.39–31.11)	<0.001	25.12 (7.29–87.80)	<0.001
*p* for trend		<0.001		<0.001	
**Dietary AFB_1_ exposure (μg/d)**					
Tertile 1 (0.00–2.33)	31/36	1.00 (reference)	–	1.00 (reference)	–
Tertile 2 (2.34–5.24)	35/32	1.32 (0.63–2.74)	0.463	1.41 (0.64–3.09)	0.391
Tertile 3 (5.25–33.62)	34/32	1.25 (0.63–2.49)	0.527	1.25 (0.58–2.68)	0.575
*p* for trend		0.593		0.648	
**Frequency of corn flour intake**					
Less than once a month	22/38	1.00 (reference)	–	1.00 (reference)	–
Once a month–less than 4 times a week	26/25	1.81 (0.87–3.78)	0.115	1.72 (0.75–3.95)	0.197
4 times a week–less than twice a day	27/19	2.65 (1.18–5.98)	0.019	3.22 (1.29–8.02)	0.012
Twice a day or more	25/18	3.26 (1.23–8.61)	0.017	3.56 (1.23–10.34)	0.019
*p* for trend		0.024		0.017	
**Mildew of stored grains**					
No	90/98	1.00 (reference)	–	1.00 (reference)	–
Yes	10/2	5.00 (1.10–22.82)	0.038	10.28 (1.44–73.22)	0.020

* Adjusted for gender, age, BMI, education level, annual income, tobacco smoking, and alcohol drinking.

**Table 3 toxins-12-00299-t003:** ORs (95% CIs) for frequency of corn flour intake with AFB_1_-related variables.

AFB_1_-Related Varibles	Frequency of Corn Flour Intake			
	Less Than Once a Month	Once a Month–Less Than 4 Times a Week	4 Times a Week–Less Than Twice a Day	Twice a Day or More
**Serum AFB_1_-Alb adduct level ***				
No. of ≤60.97/>60.97 pg/mL	36/24	25/26	24/22	15/28
Adjusted OR (95% CI) ^†^	1.00 (reference)	1.26 (0.54–2.90)	1.59 (0.68–3.69)	3.72 (1.54–9.04)
*p* value		0.594	0.282	0.004
*p* for trend	0.003			
**Dietary AFB_1_ exposure ***				
No. of ≤3.92/>3.92 μg/d	38/22	32/19	19/27	11/32
Adjusted OR (95% CI) ^†^	1.00 (reference)	1.01 (0.42–2.42)	3.04 (1.26–7.34)	7.35 (2.85–19.01)
*p* value		0.982	0.013	<0.001
*p* for trend	<0.001			

* Categorized into binary classifications of high and low levels according to the median. ^†^ Adjusted for gender, age, BMI, education level, annual income, tobacco smoking, and alcohol drinking.

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
