# Peer review of "Corn Flour Intake, Aflatoxin B1 Exposure, and Risk of Esophageal Precancerous Lesions in a High-Risk Area of Huai’an, China: A Case-Control Study"

_toxins, 2020, doi:10.3390/toxins12050299_

Round 1

Reviewer 1 Report

Dear author(s),

your paper is very interesting. For people who are not as familiar with statistics as you more expalanation on OR and trend analysis would be fine.

Best regards, the reviewer

Author Response

  1. your paper is very interesting. For people who are not as familiar with statistics as you more explanation on OR and trend analysis would be fine.
  • Response: Thank you for your comment and suggestion. In page 9, line 300-308, we have added more explanation on OR and trend analysis, as well as the 95% CI: " The obtained p for trend is used to test whether there is a certain linear trend between the independent and dependent variables."; "the OR represents the odds that an outcome will occur given a particular exposure when compared to the odds of the outcome occurring in the absence of the exposure (OR > 1, exposure is associated with higher odds of outcome; OR ˂ 1, exposure is associated with lower odds of outcome; OR = 1, exposure does not influence odds of outcome), and the 95% CI represents the precision of the OR (in practice, the 95% CI is a proxy for the presence of statistical significance if it does not overlap the null value, e.g., OR = 1)". For these explanations, a related reference is added as well ( Szumilas M. Explaining odds ratios. J Can Acad Child Adolesc Psychiatry. 2010;19(3):227-9.).

Reviewer 2 Report

Esteemed Authors,

It has been a great honor, as well as a pleasantly challenging activity, to review the article entitled Corn Flour Intake, Aflatoxin B1 Exposure, and Risk of Esophageal Precancerous Lesions in a High-risk Area of Huai'an, China: A Case-control Study.

Aflatoxins are mycotoxins produced by two species of Aspergillus, a fungus found mainly in areas with hot and humid climates. As aflatoxins are known to be genotoxic and carcinogenic, exposure through food should be kept as low as possible. Aflatoxins can occur in foods such as groundnuts, tree nuts, maize, rice, figs, and other dried grains, spices, crude vegetable oils, and cocoa beans, as a result of fungal contamination before and after harvest. Several types of aflatoxins are produced naturally. Aflatoxin B1 is the most common in food and among the most potent genotoxic and carcinogenic aflatoxins. It is produced both by Aspergillus flavus and Aspergillus parasiticus. Aflatoxin M1 is a major metabolite of aflatoxin B1 in humans and animals, which may be present in milk from animals fed with aflatoxin B1 contaminated feed. Climate change is expected to have an impact on the presence of aflatoxins in food in Europe and worldwide.

Mycotoxin levels in food usually do not produce an acute adverse effect on consumers. Still, chronic exposure may pose a significant risk to consumers who are eating these products frequently, in particular for aflatoxins. Aflatoxin B1 is a carcinogenic and genotoxic substance, for which there is no real safe level of intake. For this reason, the ALARA principle is applied, and the legal limit enforced is as low as reasonably achievable. Despite the measures taken, mycotoxins, and especially aflatoxins, remain at the top of the dangers identified and notified within the European Rapid Alert System for Food and Feed (RASFF), with a total number of 569 notifications during 2018 (the year for which there are the most recent data).

To assist the competent authorities on the precise control of aflatoxin contamination in food products, which are subject to Commission Implementing Regulation (EU) 2019/1793, a framework document was drawn up, and all Member States are guided accordingly. This document entitled "Guidance document for competent authorities for the control of compliance with EU legislation on aflatoxins" has been elaborated and is also applicable for the control of aflatoxins in food products not subject to the safeguard Regulation. This guidance document is currently reviewed to be adapted further to the application as of 14 December 2019 of Regulation (EU) 2017/625 of the European Parliament and the Council and of Implementing Regulation (EU) 2019/1793.

Practically and historically, the first serious approach to this problem was made in 2004, year in which the European Food Safety Authority (EFSA’s) CONTAM Panel adopts an opinion related to aflatoxin B1 as an undesirable substance in animal feed. The CONTAM Panel concludes that the current maximum levels of aflatoxin B1 in animal feed not only provides an adequate protection from adverse health effects in target animal species, but also prevents undesirable concentration of the metabolite aflatoxin M1 in milk. Among its recommendations, the Panel encourages monitoring of the presence of aflatoxin B1 in imported feedstuffs and aflatoxin M1 in dairy milk.

The next moment considered very important is the year 2007, the year in which EFSA assesses the possibility of a potential increase in consumers’ health risks if higher levels of aflatoxins were permitted for almonds, hazelnuts and pistachios. Increasing the current EU maximum levels of 4 µg/kg total aflatoxins in these three nuts to 8 or 10 µg/kg total aflatoxins would have minor effects on the estimated dietary exposure on the risk of cancer and the calculated margin of exposure. The Panel also concludes that exposure to aflatoxins from all food sources should be kept as low as reasonably achievable because aflatoxins are genotoxic and carcinogenic.

From this point of view, the paper is of high value due to its original character, and it treats a specific subject that is of high interest for the domain of food chemistry, food safety, toxicology, and public health. With some minor exceptions (which refers to some descriptions necessary), all materials and methods are specified and described adequately.

The article is structured following the classic model for this type of material (Research Article), comprising five parts: Introduction, Results, Discussion, Conclusions, and Materials and Methods. All the five major components of the article are balanced dimension-wise and are presented coherently and logically, tightly linked to one another. Unfortunately, the chapter on Materials and Methods is not appropriately placed: as a rule, this chapter follows immediately after the introduction and before the section devoted to discussions.

With some exceptions, all materials and methods are specified and described adequately. All iconographic documents – three tables, and three figures - were given accurate descriptions, the results were described in great detail, and the conclusions are adequate.

The list of bibliographic references is generous, the documentation is appropriate regarding the titles consulted, and all the authors are cited in the text of the paper, without exception.

The provided scientific results are exact and precise. The goal of the conducted research is well specified and delineated. The working protocol is appropriate, and the used analysis methods are correlated with the proposed objectives.

I would advise the authors to be more careful concerning the bibliography: it is preferred that the cited authors be mentioned in alphabetical order, and references without specified authors are mentioned at the end of the list of references, in chronological order. I also recommend using a single system not only in citations but also when it comes to the journals. I am referring here mainly to mentioning the following elements for each article consulted: journal, volume, issue, and pages (the DOI may also be mentioned, should the authors so desire, but the essential descriptive elements are the previously mentioned ones).

For example: page 9, lines 295-296, number 7 in the bibliographic references list: Strosnider H., Azziz-Baumgartner E., Banziger M., Bhat R.V., Breiman R., Brune M.N., DeCock K., Dilley A., Groopman J., Hell K., Henry S.H., Jeffers D., Jolly C., Jolly P., Kibata G.N., Lewis L., Liu X., Luber G., McCoy L., Mensah P., Miraglia M., Misore A., Njapau H., Ong C.N., Onsongo M.T.K., Page S.W., Park D., Patel M., Phillips T., Pineiro M., Pronczuk J., Schurz Rogers H., Rubin C., Sabino M., Schaafsma A., Shephard G., Stroka J., Wild C., Williams J.T., Wilson D. Workgroup Report: Public Health Strategies for Reducing Aflatoxin Exposure in Developing Countries. Environmental Health Perspectives (or JCR Abbreviation – Environ. Health Persp.), 2006; 114, 12, 1898-1903; DOI: https://doi.org/10.1289/ehp.9302.

Moreover, I suggest the authors to consult and include the following works in the bibliographic references list:

Bondoc I.. European Regulation in the Veterinary Sanitary and Food Safety Area, a Component of the European Policies on the Safety of Food Products and the Protection of Consumer Interests: A 2007 Retrospective. Part One: the Role of European Institutions in Laying Down and Passing Laws Specific to the Veterinary Sanitary and Food Safety Area. Universul Juridic, Supliment, 2016, pp. 12-15 (Available online: http://revista.universuljuridic.ro/supliment/european-regulation-veterinary-sanitary-food-safety-area-component-european-policies-safety-food-products-protection-consumer-interests-2007-retrospective/).

Bondoc I.. European Regulation in the Veterinary Sanitary and Food Safety Area, a Component of the European Policies on the Safety of Food Products and the Protection of Consumer Interests: A 2007 Retrospective. Part Two: Regulations. Universul Juridic, Supliment, 2016, pp. 16-19 (Available online: http://revista.universuljuridic.ro/supliment/european-regulation-veterinary-sanitary-food-safety-area-component-european-policies-safety-food-products-protection-consumer-interests-2007-retrospective-2/).

All these papers approach the matter of food safety legislation enforced within the European Union, which usually constitutes a blueprint for the law in third countries. The documents mentioned outline the European legislative environment, starting with the year 2007, the year of the penultimate geo-political enlargement of the European Union. I want to add that all four recommended papers have been indexed in CAB International and HeinOnline, the largest and most extensive worldwide database for documents in the legal field.

The two articles recommended represent a systematic database regarding all the normative acts issued and applicable at the European Union level in 2007, in the field of food safety, including the regulations on the risks represented by mycotoxins.

The obtained results are analyzed and interpreted correctly, and their practical value is visible.

As for the grammar of the paper, the article is very well written: only a few shortcomings in the grammar of the text can be mentioned, as follows:

Page 1, line 21 – replace “of esophageal” with “of the esophageal”;

Page 1, line 38 – replace “among” with “between”;

Page 2, line 56 – replace “of biologically” with “for the biologically”;

Page 2, line 56 – replace “has relatively” with “has a relatively”;

Page 3, line 86 – replace “between two” with “between the two”;

Page 3, line 91 - replace “between two” with “between the two”;

Page 5, line 114 – replace “alcoholic” with “alcohol”;

Page 6, line 136 – replace “of hepatocellular” with “of the hepatocellular”;

Page 6, line 142 – replace “areas than low” with “areas rather than the low”;

Page 6, line 162 – replace “in loss” with “in the loss”;

Page 6, line 173 – replace “is a complex” with “are a complex”;

Page 7, line 221 – replace “invasion” with “the invasion”;

Page 7, line 228 – replace “dietary” with “the dietary”;

Page 8, line 232 – replace “on three” with “in three”;

Page 8, line 239 – replace “for test” with “for the test”;

Page 8, line 244 – replace “filtrate” with “the filtrate”;

Page 8, line 245 - replace “for test” with “for the test”;

Page 8, line 259 – replace “questionnaire” with “the questionnaire”;

Page 8, line 272 – replace “for linear” with “for a linear”.

As a general conclusion regarding the grammar, the text does not contains other mistakes that need to be corrected. As for the editing (writing), part is concerned, the work should be checked once again carefully.

The article itself, like any other article, has certain improvable aspects. By these aspects, I mean the major constituting parts of the article, but also some elements that are related to details or writing – for example, the use of abbreviations only after mention in extenso; a more accurate description of some chemicals; more accuracy when it comes to expressing some concentrations, etc.. However, the material as a whole, can be considered necessary for academic staff, for researchers in the field and even for the broad public.

Together with other positive elements, the scientific relevance and quality of the presentation will surely make the article attractive to a broad audience, and especially to the authors interested in the fields of food chemistry, toxicology, food analysis, food residues, risk assessment, food safety, and public health.

Provided that the authors revise the material and improve on the elements mentioned above, the paper may be published in the Toxins.

            Best Regards,

            Reviewer

Author Response

  1. The article is structured following the classic model for this type of material (Research Article), comprising five parts: Introduction, Results, Discussion, Conclusions, and Materials and Methods. All the five major components of the article are balanced dimension-wise and are presented coherently and logically, tightly linked to one another. Unfortunately, the chapter on Materials and Methods is not appropriately placed: as a rule, this chapter follows immediately after the introduction and before the section devoted to discussions.
  • Response: Thank you for your work on this manuscript. We really appreciate the detailed information that you share with us, especially the part of regulations which related to mycotoxins. For the chapter on Materials and Methods, we used the template provided by Toxins, and we also noticed that all the articles published in Toxins would place this chapter after Conclusions. Thank you for your reminder.

  1. I would advise the authors to be more careful concerning the bibliography: it is preferred that the cited authors be mentioned in alphabetical order, and references without specified authors are mentioned at the end of the list of references, in chronological order. I also recommend using a single system not only in citations but also when it comes to the journals. I am referring here mainly to mentioning the following elements for each article consulted: journal, volume, issue, and pages (the DOI may also be mentioned, should the authors so desire, but the essential descriptive elements are the previously mentioned ones).

For example: page 9, lines 295-296, number 7 in the bibliographic references list: Strosnider H., Azziz-Baumgartner E., Banziger M., Bhat R.V., Breiman R., Brune M.N., DeCock K., Dilley A., Groopman J., Hell K., Henry S.H., Jeffers D., Jolly C., Jolly P., Kibata G.N., Lewis L., Liu X., Luber G., McCoy L., Mensah P., Miraglia M., Misore A., Njapau H., Ong C.N., Onsongo M.T.K., Page S.W., Park D., Patel M., Phillips T., Pineiro M., Pronczuk J., Schurz Rogers H., Rubin C., Sabino M., Schaafsma A., Shephard G., Stroka J., Wild C., Williams J.T., Wilson D. Workgroup Report: Public Health Strategies for Reducing Aflatoxin Exposure in Developing Countries. Environmental Health Perspectives (or JCR Abbreviation – Environ. Health Persp.), 2006; 114, 12, 1898-1903; DOI: https://doi.org/10.1289/ehp.9302.

  • Response: I know what you mean about the cited authors could be mentioned in alphabetical order. Sometimes we use this reference style as you said, but we selected the required style provided by Toxins this time. In addition, we downloaded the paper published in Toxins, and found that all the authors should be listed if the number of authors is 10 or less. So we have made some changes based on the Toxins journal's style. For the "page 9, lines 295-296, number 7 in the bibliographic references list", we follow the your suggestion and correct it now. It can be found in page 10, lines 330-332, number 7 in the current version of manuscript.

  1. Moreover, I suggest the authors to consult and include the following works in the bibliographic references list:

Bondoc I.. European Regulation in the Veterinary Sanitary and Food Safety Area, a Component of the European Policies on the Safety of Food Products and the Protection of Consumer Interests: A 2007 Retrospective. Part One: the Role of European Institutions in Laying Down and Passing Laws Specific to the Veterinary Sanitary and Food Safety Area. Universul JuridicSupliment, 2016, pp. 12-15 (Available online: http://revista.universuljuridic.ro/supliment/european-regulation-veterinary-sanitary-food-safety-area-component-european-policies-safety-food-products-protection-consumer-interests-2007-retrospective/).

Bondoc I.. European Regulation in the Veterinary Sanitary and Food Safety Area, a Component of the European Policies on the Safety of Food Products and the Protection of Consumer Interests: A 2007 Retrospective. Part Two: Regulations. Universul Juridic, Supliment2016, pp. 16-19 (Available online: http://revista.universuljuridic.ro/supliment/european-regulation-veterinary-sanitary-food-safety-area-component-european-policies-safety-food-products-protection-consumer-interests-2007-retrospective-2/).

  • Response: Thank you for your suggestion. We add these references and related contents in the Introduction, page 2, lines 61-65: "At the same time, the veterinary sanitary and food safety area was subjected a significant legislative pressure with a large amount of passed European acts, including the regulations on the risks represented by mycotoxins[9]. All the characteristics make regulations extremely efficient work tools, with a widespread use in regulating the most varied subdomains, from animal health issues to contaminants and mycotoxins[10]. " In addition, based on your comments, we also add a reference related to regulation here, page 2, lines 57-61: "In 2007, European Food Safety Authority (EFSA) assessed the possibility of potential increase in consumers' health risks if higher levels of aflatoxins were permitted for almonds, hazelnuts and pistachios. The Panel also concluded that exposure to aflatoxins from all food sources should be kept as low as reasonably achievable because aflatoxins are genotoxic and carcinogenic[8]." ( European Food Safety Authority. Effects on public health of an increase of the levels for aflatoxin total from 4 µg/kg to 10 µg/kg for tree nuts other than almonds, hazelnuts and pistachios - Statement of the Panel on Contaminants in the Food Chain. Efsa J. 2009;7(6):1168.) The three added references can be found in page 10, lines 333-356, number 8-10.

  1. As for the grammar of the paper, the article is very well written: only a few shortcomings in the grammar of the text can be mentioned, as follows:

Page 1, line 21 – replace “of esophageal” with “of the esophageal”;

Page 1, line 38 – replace “among” with “between”;

Page 2, line 56 – replace “of biologically” with “for the biologically”;

Page 2, line 56 – replace “has relatively” with “has a relatively”;

Page 3, line 86 – replace “between two” with “between the two”;

Page 3, line 91 - replace “between two” with “between the two”;

Page 5, line 114 – replace “alcoholic” with “alcohol”;

Page 6, line 136 – replace “of hepatocellular” with “of the hepatocellular”;

Page 6, line 142 – replace “areas than low” with “areas rather than the low”;

Page 6, line 162 – replace “in loss” with “in the loss”;

Page 6, line 173 – replace “is a complex” with “are a complex”;

Page 7, line 221 – replace “invasion” with “the invasion”;

Page 7, line 228 – replace “dietary” with “the dietary”;

Page 8, line 232 – replace “on three” with “in three”;

Page 8, line 239 – replace “for test” with “for the test”;

Page 8, line 244 – replace “filtrate” with “the filtrate”;

Page 8, line 245 - replace “for test” with “for the test”;

Page 8, line 259 – replace “questionnaire” with “the questionnaire”;

Page 8, line 272 – replace “for linear” with “for a linear”.

  • Response: Thank you very much for these detailed corrections. The replacements have been made and tracked in the manuscript.

Reviewer 3 Report

I think that this article is well written. The number of cases used are enough for this study. The statistical part is  well used for this study. However the conclusions are too short and in my opinion must be better written and underlined.

Author Response

  1. I think that this article is well written. The number of cases used are enough for this study. The statistical part is well used for this study. However the conclusions are too short and in my opinion must be better written and underlined.
  • Response: Thank you for your comments. The chapter of conclusions was too short and thus we add more important findings in the conclusions now: "Corn flour intake was positively associated with serum biomarker for AFB1 (AFB1-Alb adduct level), dietary AFB1 exposure and the risk of EPL, while the increased level of serum AFB1-Alb adduct and mildew of stored grains were significantly associated with the increased risk of EPL. Therefore, it can be concluded that corn flour is likely to be an essential source of AFB1 in Huai'an District due to the suitable environment for fungal growth during the grain storage"; and "In addition, some individuals may be more susceptible to AFB1 in some case, such as enzymatic capacity and gene regulation which may affect the metabolism of AFB1 in vivo, contribute to the increased level of serum AFB1 exposure biomarker, and result in the increased risk of tumor initiation." The improved Conclusions can be found in page 7, lines 207-218 now.